Halichoeres sanchezi n. sp., a new wrasse from the Revillagigedo Archipelago of Mexico, tropical eastern Pacific Ocean (Teleostei: Labridae)

http://orcid.org/0000-0002-8728-9585 Victor Benjamin C. 1 2 ben@coralreeffish.com
Frable Benjamin W. 3
http://orcid.org/0000-0002-0599-9699 Ludt William B. 4
1 Guy Harvey Research Institute, Nova Southeastern University , Dania Beach, Florida , United States
2 Marine Biology, Ocean Science Foundation , Irvine, California , United States
3 Marine Vertebrate Collection, Scripps Institution of Oceanography , La Jolla, California , United States
4 Ichthyology, Natural History Museum of Los Angeles County , Los Angeles, California , United States
Venmathi Maran Balu Alagar
Electronic publication date: 2024 Feb 28
Publication date: 2024
Volume: 12
Electronic Location ID: e16828
Received 2023 Oct 31; Accepted 2024 Jan 4
Copyright: © 2024 Victor et al.
Copyright year: 2024
Copyright holder: Victor et al.
License: This is an open access article distributed under the terms of the Creative Commons Attribution License, which permits unrestricted use, distribution, reproduction and adaptation in any medium and for any purpose provided that it is properly attributed. For attribution, the original author(s), title, publication source (PeerJ) and either DOI or URL of the article must be cited.
License URL: https://creativecommons.org/licenses/by/4.0/

Keywords: Ichthyology, Taxonomy, Biogeography, Endemism, Coral reef fishes, Eastern Pacific, New species, Mexico, DNA barcoding

Funding: SIO Director’s Office, the Scripps Oceanographic Collections Fund, and the Lehman-Kennel Endowment for the Collections at SIO The Natural History Museum of Los Angeles County supported the expedition The East Pacific Corridor Alliance supported the expedition National Science Foundation supported the µCT scanner at the NHMLA DBI 2215184 Participation by Benjamin W. Frable was supported by the SIO Director’s Office, the Scripps Oceanographic Collections Fund, and the Lehman-Kennel Endowment for the Collections at SIO. The Natural History Museum of Los Angeles County supported the expedition. The East Pacific Corridor Alliance supported the expedition. Funding from the National Science Foundation supported the µCT scanner at the NHMLA (DBI 2215184). The funders had no role in study design, data collection and analysis, decision to publish, or preparation of the manuscript.

==============================
A new labrid fish species, Halichoeres sanchezi n. sp., is described from eight specimens collected in the Revillagigedo Archipelago in the tropical eastern Pacific Ocean, off the coast of Mexico. The new species belongs to the Halichoeres melanotis species complex that is found throughout the region, differing by 2.4% in the mtDNA cytochrome c oxidase I sequence from its nearest relative, H. melanotis from Panama, and 2.9% from Halichoeres salmofasciatus from Cocos Island, off Costa Rica. The complex is distinguished from others in the region by having a black spot on the opercular flap and a prominent black area on the caudal fin of males. The juveniles and initial phase of the new species closely resemble those of H. salmofasciatus and Halichoeres malpelo from Malpelo Island of Colombia, differing in having an oblong black spot with a yellow dorsal margin on the mid-dorsal fin of initial-phase adults as well as on juveniles. In contrast, the terminal-phase male color pattern is distinct from other relatives, being vermilion to orangish brown with dark scale outlines, a white patch on the upper abdomen, and a prominent black band covering the posterior caudal peduncle and base of the caudal fin. The new species adds to the list of endemic fish species for the isolated archipelago and is an interesting case of island endemism in the region. The discovery was made during the joint 2022 collecting expedition to the archipelago, which featured a pioneering collaborative approach to an inventory of an island ichthyofauna, specifically including expert underwater photographers systematically documenting specimens in situ, before hand-collection, and then photographed fresh, tissue-sampled, and subsequently vouchered in museum collections.

Introduction

The geography of the tropical eastern Pacific Ocean (TEP) provides a useful proving ground for answering questions about marine island biogeography. There is a long and relatively linear coastline with a narrow continental shelf and a series of isolated offshore islands. Unlike most other tropical marine regions, each island group contains a different set of the regional shorefish fauna (Allen & Robertson, 1994) and the patterns of occurrence, especially the prevalence of endemism and degrees of connectivity, can be directly assessed and hypotheses can be tested. The Revillagigedo Archipelago of Mexico, forming the Reserva de la Biosfera Archipielago de Revillagigedo, includes the islands of Clarion, Socorro, San Benedicto, and Roca Partida, and is about 400 km south of the Baja California peninsula and about 700 km west of Manzanillo. It is one of the more remote and least visited of the island groups of the TEP, with a short history of collecting expeditions and a very limited number of museum collections, especially compared to more well-surveyed TEP islands, such as the Galapagos Archipelago.

In November 2022, a joint expedition formed by an international team of ichthyologists and a cadre of experienced underwater photographers, visited the four islands of the archipelago and compiled a comprehensive inventory of the shallow shorefish fauna, with in situ photographic documentation of all of the fish species encountered. All species collected were sampled for tissues and subsequently DNA-barcoded, i.e., sequenced for the mtDNA cytochrome c oxidase I gene (COI). One of the more compelling aspects of the expedition was the contemporaneous underwater documentation, including on digital video, of the discovery and collection of the holotype of a new species of fish, in this case a new labrid species of the large genus Halichoeres Ruppell, 1835.

The new species is the local representative of the widespread TEP Halichoeres melanotis (Gilbert, 1890) species complex, comprising H. melanotis along the mainland, Halichoeres salmofasciatus Allen & Robertson, 2002, endemic to Cocos Island, and Halichoeres malpelo Allen & Robertson, 1992, endemic to Malpelo Island (Robertson & Allen, 2015). The new species was not listed in recent species lists for the Revillagigedo Archipelago (Del Moral-Flores, Gracian-Negrete & Guzman-Camacho, 2016; Fourriere et al., 2016). It had been photographed at least once previously, but had not been identified with any certainty. It is a particularly elusive wrasse, with adults difficult to approach, accounting for the dearth of documentation despite many tourist divers photographing fishes in the archipelago. We collected a series of specimens on San Benedicto, the northernmost island of the Revillagigedo Archipelago (Fig. 1).

Figure 1 Type location of Halichoeres sanchezi n. sp. at El Canyon dive site in the Caletilla Volteadura bay at the south side of San Benedicto Island, Revillagigedo Archipelago, Colima, Mexico. Photo credit: Raphael Gatti, https://commons.wikimedia.org/wiki/File:San_Bendicto_from_above.jpg, licensed under a Creative Commons Attribution 4.0 International (CC BY 4.0)).

Materials and Methods

Specimens of the new species are accessioned in the Coleccion Nacional de Peces at the Universidad Nacional Autonoma de Mexico (CNP-IBUNAM), the Natural History Museum of Los Angeles County (LACM), and the Marine Vertebrate Collection of Scripps Institution Oceanography (SIO). Permits for the travel, boats, collection, and export were issued to the Universidad Nacional Aut6noma de Mexico by the Secretariat of Agriculture, Livestock, Rural Development, Fisheries and Food (SAGARPA) of the Federal Government of Mexico (permit number PPF/DGOPA-A-099/22). Measurements and counts of congeners used for comparative purposes were from specimens housed at LACM, SIO, and the NMNH or values taken from the literature (Allen & Robertson, 1992, 1994, 2002; Robertson & Allen, 2015).

Measurements were taken as previously described in Victor (2016). Specifically, the length of specimens is given as standard length (SL), measured from the median anterior end of the upper lip to the base of the caudal fin (posterior end of the hypural plate); body depth is the greatest depth from the base of the dorsal-fin spines to the ventral edge of the abdomen (correcting for any malformation of preservation); body width is measured just posterior to the gill opening; head length from the front of the upper lip to the posterior end of the opercular flap; orbit diameter is the greatest fleshy diameter of the orbital rim, and interorbital width the least bony width between the orbits; snout length is measured from the median anterior point of the upper lip to the nearest fleshy rim of the orbit; caudal-peduncle depth is the least depth, and caudal-peduncle length the horizontal distance between verticals at the rear base of the anal fin and the caudal-fin base; predorsal, prepelvic and preanal lengths are oblique measurements taken from the median anterior point of the upper lip to the base of the first spine of each respective fin; lengths of spines and rays are measured to their extreme bases; caudal-fin and pectoral-fin lengths are the length of the longest ray; pelvic-fin length is measured from the base of the pelvic spine to the tip of the longest soft ray. The upper rudimentary pectoral-fin ray is included in the count. Lateral-line scale counts list the last pored scale that overlaps the end of the hypural plate as +1. The count of gill rakers is made on the first gill arch and includes all rudiments. Proportional measurements were taken with digital calipers or from radiograph images using ImageJ (https://imagej.net/) and are rounded to the nearest 0.1. Counts were taken from specimens, digital radiographs, and micro-computer tomography (microCT) scans. Counts and measurements for the paratypes are shown in parentheses following data for the holotype (some counts and measurements were not made on the juvenile paratype CNP-IBUNAM 23979). Proportional morphological measurements as percentages of the standard length are presented in Table 1 and proportional morphological measurements in the diagnosis and description are presented as the number of times in SL, HL, or body depth.

Table 1 Proportional measurements of type specimens of Halichoeres sanchezi n. sp., as percentages of standard length.

	holotype	paratypes	
	LACM
61379	LACM 61380	SIO 23-50	CNP IBUNAM
23979	SIO 23-51	SIO 23-48	CNP IBUNAM
23979	
Standard length (mm)	123.1	53.3	46.2	40.4	39.1	36.5	32.3	
Body depth	27.8	25	24.5	23.8	24.2	24.9	26.4	
Body width	11.3	9.4	11.3	9.8	10.7	10.6	9.3	
Head length	30.8	30.9	32	32.7	31.6	31.5	32.5	
Snout length	8.8	8.5	9.1	8.2	8.1	8.5	8.2	
Orbit diameter	5.6	6.8	6.5	7.6	7.5	7.8	7.9	
Interorbital width	5.9	5.5	5.7	5.5	4	4.1	4.2	
Upper-jaw length	7.2	5.8	6.1	6.4	6.9	6.5	6.6	
Caudal-peduncle depth	14.2	13.7	13.2	13.4	13.5	13.2	13.2	
Caudal-peduncle length	12.7	7.7	9.0	8.4	8.5	6.6	8.9	
Predorsal length	31.5	29.8	29.9	29.4	29.9	29.7	29.6	
Preanal length	47.4	53.5	55.6	58	57.1	55.3	56.5	
Prepelvic length	27.5	32.1	33.3	32.9	30.9	32.7	30.9	
Dorsal-fin base	66.8	60.2	62.9	62.4	64.6	66.7	66.8	
First dorsal-fin spine	5.3	–	4.1	5.0	3.9	3.7	3.8	
Longest dorsal-fin spine	8.2	8.2	8.2	9.2	8.4	8.5	8.9	
Longest dorsal-fin ray	12.5	10.5	10.8	10.5	10.5	10.6	11.2	
Anal-fin base	39.0	39.3	35.5	37.3	37.9	35.4	33.9	
First anal-fin spine	2.9	2.3	1.9	2.2	2.4	2.2	2.1	
Second anal-fin spine	4.9	5.6	5.4	6.1	6.3	6.2	5.9	
Third anal-fin spine	6.8	7.3	7.6	7.9	7.6	7.6	7.2	
Longest anal-fin ray	10.4	9.6	9.3	9.8	9.9	9.9	9.6	
Caudal-fin length	16.4	19.9	21.7	23.3	24.9	24.5	25.8	
Pectoral-fin length	17.9	17.2	16.2	19.1	17.6	17.8	18.1	
Pelvic-fin spine length	8.8	8.8	8.6	9.3	9.2	8.1	7.8	
Pelvic-fin length	13.3	13.1	13.1	13.0	13.4	13.2	13.4	

Fluorescent emission was documented using a Canon 5D Mark IV DSLR camera with a Canon EF 100 mm f/2.8L Macro IS USM lens. Photographs were taken using two NightSea Xite Fluorescent Flashlights that emitted royal blue light (ca 440–460 nm). Each light was held at approximately 45° to the subject and was equipped with a 15° diffuser cap. A 500 nm Tiffin #12 Yellow Filter was placed in front of the lens for all photographs. Micro-computed tomography (µCT) scans were used to visualize internal skeletal anatomy. Scans were performed at the NHMLA on a Bruker Skyscan 1273. Scans of the holotype and one of the paratypes were conducted with and without a 1 mm aluminum filter, with 70–90 kV, 166–214 µA, 471–636 ms exposure times, with a 10–34 µm voxel size. All scans were reconstructed using the software NRecon v2.2.0.6 (Bruker, Billerica, MA, USA) and visualized using CTVox (Bruker, Billerica, MA, USA). CT scans were uploaded to Morphosource.org for the holotype (https://www.morphosource.org/concern/media/000570594?locale=en) and the largest paratype (https://www.morphosource.org/concern/media/000570600?locale=en).

A sample of muscle tissue from the holotype was preserved in 90% ethanol and maintained at room temperature until sequencing. Standard DNA extractions, primers, and sequencing protocols followed those described in Victor, Alfaro & Sorenson (2013). Specimen information and barcode-sequence data from this study were compiled using the Barcode of Life Data Systems (BOLD) (Ratnasingham & Hebert, 2007) and all sequences are publicly accessible on BOLD and GenBank. Sequence divergences were calculated using BOLD with the Kimura 2-parameter (K2P) model. Genetic distances were calculated by the BOLD algorithm, both as uncorrected p-distances and as K2P distances. The COI sequence for the new species was compared to other members of the H. melanotis species complex as well as to the remaining publically available New World Halichoeres species. The genus is well covered in both the TEP and western Atlantic, with almost all species presently sequenced for COI. The unrelated Indo-Pacific species Halichoeres argus (Bloch & Schneider, 1801) was selected as an outgroup. A consensus neighbor-joining phylogenetic tree was constructed in Geneious Prime v2023.2.1 using the Tamura-Nei genetic distance model and 100 bootstrap replicates.

The electronic version of this article in Portable Document Format (PDF) will represent a published work according to the International Commission on Zoological Nomenclature (ICZN), and hence the new names contained in the electronic version are effectively published under that Code from the electronic edition alone. This published work and the nomenclatural acts it contains have been registered in ZooBank, the online registration system for the ICZN. The ZooBank LSIDs (Life Science Identifiers) can be resolved and the associated information viewed through any standard web browser by appending the LSID to the prefix http://zoobank.org/. The LSID for this publication is urn:lsid:zoobank.org:pub:0FF79EE8-EB5A-45B3-98CE-9A420881DAA0. The online version of this work is archived and available from the following digital repositories: PeerJ, PubMed Central SCIE and CLOCKSS.

Halichoeres sanchezi n. sp. urn:lsid:zoobank.org:act:608591F4-2A1B-4271-9BB2-0F0B2BD2DC4E

Tailspot Wrasse, Doncella Colimanchada (Spanish)

mtDNA COI sequence BIN https://doi.org/10.5883/BOLD:AFA9780

Figures 2–6, Table 1.

Figure 2 Halichoeres sanchezi n. sp., type specimens.

(A) Terminal-phase (TP) male holotype, LACM 61379, 123.1 mm SL. (B) Initial-phase (IP) paratype, LACM 61380, 46.2 mm SL. Photo credit: Alasdair Dunlap-Smith.

Figure 3 Halichoeres sanchezi n. sp., first known underwater photographs, from Socorro, Revillagigedo Archipelago in 2013.

(A) Small IP. (B) IP adult. (C) Two TP males. Photo credit: Kreg Martin.

Figure 4 Halichoeres sanchezi n. sp., TP males underwater, San Benedicto, Revillagigedo Archipelago, Colima, Mexico.

(A) Photo credit: Sara Richter. (B) Image adjusted to black-and-white due to color aberration in original. Photo credit: Andres Hernandez.

Figure 5 Halichoeres sanchezi n. sp., juvenile and IP adults underwater, San Benedicto, Revillagigedo Archipelago, Colima, Mexico.

(A) Small juvenile. Photo credit: Allison Morgan Estape (B) Large juvenile. Photo credit: Jeff Haines. (C) IP adult. Photo credit: Allison Morgan Estape.

Figure 6 Halichoeres sanchezi n. sp., type specimens: fresh, with fluorescence, and preserved.

Holotype, at left, LACM 61379, 123.1 mm SL: (A) Fresh. Photo credit: Alasdair Dunlap-Smith. (B) With fluorescence and (C) preserved. Photo credit: William B. Ludt. Paratype, at right, LACM 61380, 53.3 mm SL: (A1) Fresh. Photo credit: Alasdair Dunlap-Smith. (B1) With fluorescence and (C1) preserved. Photo credit: William B. Ludt.

Holotype. LACM 61379, 123.1 mm SL, Mexico, Colima, Revillagigedo Archipelago, San Benedicto Island, canyon rubble plain site, sta. WBL 275, 19°17.740′N, 110°48.466′W (19.2957, −110.8078); 21–22 m depth; spear, W.B. Ludt, 30 November 2022.

Paratypes. (Seven specimens 32.3–53.3 mm SL), CNP-IBUNAM 23979, 39.1 mm SL, sta. WBL 274, same locality as holotype, quinaldine and handnet, B.W. Frable, 30 November 2022; CNP-IBUNAM 23979, 2, 19.8–32.3 mm SL, sta. WBL 274, same locality as holotype, quinaldine and handnet, C.A. Sanchez et al., 30 November 2022; LACM 61380, 53.3 mm SL, sta. WBL 274, same locality as holotype; quinaldine and handnet, B.W. Frable, 30 November 2022; SIO 23-48, 40.4 mm SL, sta. WBL 279, 19°17.795′N, 110°48.351′W, quinaldine and handnet, C.A. Sanchez, 30 November 2022; SIO 23-50, 46.2 mm SL, sta. WBL 274, same locality as holotype, quinaldine and handnet, C.A. Sanchez et al., 30 November 2022; SIO 23-51, 36.5 mm SL, sta. WBL 275, same locality as holotype; quinaldine and handnet, B.W. Frable, 30 November 202.

Diagnosis. Dorsal-fin rays IX,12; anal-fin rays III,12; pectoral-fin rays 13; lateral-line pored scales 26 (+1 on caudal-fin base); suborbital pores 6; gill rakers 14–16; a pair of large, projecting, and slightly recurved canine teeth anteriorly in each jaw, the lowers curving forward and fitting between uppers when mouth closed, a second pair of canines about half size of first, followed by rows of mostly caniniform teeth, no canine posteriorly at corner of mouth; body elongate, depth 3.6–4.2 in SL; body width 2.2–3.0 in depth; caudal fin slightly rounded; TP vermilion on head to orangish-brown on body, yellow-cream ventrally with a dark cross-hatch pattern outlining scales, opercular flap diffusely dark and purple; a prominent large black blotch covering rear caudal peduncle and proximal half of caudal fin, a pale patch over mid-abdominal area underlying pectoral fin; a small black spot on first dorsal-fin membrane; IP and juvenile fish grey ventrally becoming yellowish posteriorly with a broad midlateral red band from snout to caudal peduncle, often breaking up into horizontal block segments, including a distinct black crescent or oval spot on expanded soft flap of upper operculum, and ending in a horizontal oblong black spot at caudal-fin base; a similar narrower band runs along upper body below base of dorsal fin from snout to upper caudal peduncle; fins clear except for a large oblong black spot, edged in yellow dorsally, centered over membranes of last dorsal-fin spine and first three rays and extending partially onto upper body (present in both juvenile and IP adults).

Description. Dorsal-fin rays IX,12; anal-fin rays III,12, all soft dorsal and anal-fin segmented rays branched, last split to base; pectoral-fin rays 13, first rudimentary, second unbranched, all remaining rays branched; pelvic-fin rays I,5; principal caudal-fin rays 6+6, upper procurrent rays 6, anteriormost 4 unsegmented, remaining rays segmented; lower procurrent rays 6, anteriormost 4 unsegmented, remaining rays segmented; pored lateral-line scales 26 (+1 on caudal-fin base); gill rakers 14 (14–16); branchiostegal rays 5; vertebrae 10+15.

Body elongate and compressed, depth 3.6 (3.8–4.2) in SL, body width 8.9 (8.9–10.8) in SL and 2.5 (2.2–2.8) in depth; head length 3.3 (3.0–3.2) in SL; snout pointed and short, its length 3.5 (3.5–4.1) in HL; orbit diameter 5.5 (4.1–4.9) in HL; interorbital space broadly convex, least bony width 5.2 (5.6–7.9) in HL; caudal peduncle short and narrow, least depth 2.2 (2.3–2.5) in HL, caudal-peduncle length 2.7 (3.6–4.8) in HL.

Mouth small, terminal, oblique at about 45 degrees, upper-jaw length 4.3 (4.6–6.3) in HL; two pairs of enlarged canine teeth at front of upper jaw (two canines per side, anterior pair larger than posterior pair) and two pairs of enlarged canine teeth at front of lower jaw (two canines per side, anterior pair larger than posterior pair) fitting between upper pair when mouth is closed; teeth behind enlarged canines in a regular row of 10–15 caniniform to conical teeth in each quadrant; no posterior canine at corner of mouth. Upper preopercular margin free and smooth nearly to level of lower edge of orbit; lower margin free anterior to a vertical through anterior nostril. Gill rakers short, longest on first arch (at angle) about one-quarter length of longest gill filament. Nostrils small, in front of anterior edge of orbit. Head pores comprise two over maxilla, then two anterior to orbit, followed by a curving suborbital series of six pores in a single row up to rear mid-level of orbit.

Scales thin, cycloid, and less than half as high on thorax as largest scales on flanks, still smaller ventroanteriorly; head naked except for irregular rows of small partially embedded scales on nape except midline; several progressively smaller scales on basal area of median fins and a mid-ventral scale projecting posteriorly from pelvic-fin base. Scale rows above lateral line 4 and below lateral line 8; circumpeduncular scales 16. Lateral line continuous, nearly following contour of back to about nineteenth pored scale, below base of about eighth dorsal-fin soft ray, where deflected ventrally to a straight peduncular portion; anteriormost scales in holotype with three pores per scale, after a few scales becoming two pores, and straight section with a single pore per scale; on all (smaller) paratypes only a single pore per scale, last pored scale on caudal-fin base.

Origin of dorsal fin above anterior edge of second lateral-line scale; predorsal length 3.2 (3.3–3.4) in SL; dorsal-fin base 1.5 (1.5–1.7) in SL; dorsal-fin spines progressively longer, first 5.9 (6.5–7.8) in HL; sixth dorsal-fin spine longest 3.8 (3.6–3.9) in HL; ninth dorsal-fin soft ray longest, 2.5 (2.9–3.1) in HL; origin of anal fin below base of last dorsal-fin spine; preanal length 2.1 (1.7–1.9) in SL; anal-fin base 2.6 (2.5–2.9) in SL; first anal-fin spine short, 10.6 (13.3–16.4) in HL; second anal-fin spine 6.3 (5.4–5.9) in HL; third anal-fin spine 4.5 (4.1–4.7) in HL; ninth anal-fin soft ray longest 3.0 (3.2–3.5) in HL; caudal-fin length 1.9 (1.3–1.5) in HL; second or third pectoral- fin ray usually longest, 1.6 (1.7– 2.0) in HL; pelvic-fin spine short, 3.5 (3.4–4.3) in HL, pelvic-fin length 2.3 (2.4–2.5) in HL.

Color in life. (Figs. 2–6) Fresh TP specimens have a vermilion head and orangish-brown body, grading to yellowish-cream ventrally. Head mostly uniformly colored, jaws paler, yellowish below level of mandible, with a prominently reddish orange iris and a reddish orange opercular flap with irregular indigo lines and black and purple in diffuse patches. Body with a cross-hatched pattern of rows of scales with dark brown crescents with reddish borders on each scale, grading to yellowish ventrally where spots are less conspicuous; upper abdominal area underlying pectoral fin with opaque white underlying scales forming a conspicuous white patch. Rear body with a large black patch covering end of caudal peduncle and extending over proximal half of caudal fin. Dorsal fin with a small black spot centered on first interspinous membrane and remaining dorsal-fin membranes with irregular yellowish bands and a blue marginal band along soft-dorsal-fin margin; small bright indigo spots at bases of anterior dorsal-fin spines; caudal-fin rays brownish orange fading to yellow distally, membranes translucent brown distally; anal fin yellow with bluish bands midfin and along margin; pectoral fins translucent, orange at base, axil with a small dark spot; pelvic fins with yellow band anteriorly.

Underwater, where red color is absorbed, head and body appear dusky purple to greenish, with dark cross-hatching from outlined scales; more prominent visible markings are pale lips, a large white patch over upper abdomen beneath pectoral fin, a pale dorsal-fin base with an expanded pale area just before black area on caudal peduncle, and a conspicuous black caudal blotch covering end of caudal peduncle and basal half of caudal fin.

IP fish pinkish grey dorsally grading to white on abdomen becoming yellowish posteriorly with a broad brick-red band running along lateral midline from snout to caudal peduncle, ending in a horizontal oblong black spot just above midline on caudal-fin base, iris bright red-orange in line with band, a distinct black crescent with indigo to purplish anterior margin on opercular flap; upper body with a less-prominent, thinner, reddish band running from upper snout to dorsal caudal peduncle, bands can form horizontal block segments when fish are disturbed. Dorsal fin with a large oblong black spot on basal two-thirds of fin, extending partially onto body scales, centered on last spine and first three rays, with most of upper margin of spot edged with yellow and white, forming a partial ocellus; a small dark spot in the pectoral-fin axil; median fin membranes translucent with brownish or faint purple-red rays. Juveniles (less than 20 mm SL) with similar color pattern but black spots on mid-dorsal fin and caudal-fin base relatively larger and a more rounded opercular-flap black spot.

Color under fluorescence. (Fig. 6) Entire body of TP male without fluorescent emission except for slight yellow emission on pale patch beneath pectoral fin; dorsal, anal and pelvic-fin spines and rays faint yellow. Entire body of IP fish bright red; eye dark without fluorescence except for upper half to two-thirds of iris red; small, round opercular black spot; round to oblong black spots at midpoint of dorsal fin and slightly above lateral line at base of caudal fin.

Color in alcohol. (Fig. 6) Head and body of TP male brownish with residual dark-centered scale rows and black markings on first dorsal-fin membrane and rear caudal blotch persisting in preservative; IP and juvenile individuals pale brownish with black marks on opercular flap, rounded spot on mid- dorsal fin, and at end of mid-lateral body band on caudal-fin base.

Etymology. Named for Prof. Carlos Armando Sanchez Ortiz, of the Programa de Investigacion para la Conservacion de la Fauna Arrecifal (PFA), Biologia Marina, Universidad Autonoma de Baja California Sur (UABCS) in La Paz, Baja California Sur, Mexico, in recognition of his contributions to the study of the marine communities of Pacific Mexico and who organized the 2022 expedition and collected the first specimen of the new species.

Distribution. The new species is presently known only from the Revillagigedo Archipelago: it has been documented on Socorro Island by the underwater photographs of Kreg Martin in 2013 (Fig. 3) and from San Benedicto Island by our 2022 expedition. No other records exist in the compendium of underwater photographs we have reviewed or in museum collections we have examined.

Habitat. Specimens were only observed and collected at a specific site off the southern end of San Benedicto, just west of the popular El Canyon dive site at Caletilla Volteadura (Fig. 1). The site was a large, even plain, approximately 21–22 m deep, composed of volcanic gravel-rubble surrounded by lava boulders and with a few boulders dispersed in the plain amongst a dark bottom. Patches of white gravel were interspersed among the dark volcanic gravel-rubble areas. Juveniles and smaller IP fish were found generally just off the bottom around boulders and over lighter patches of gravel while larger IP and TP individuals were observed slightly higher in the water column around the boulders on the eastern edge of the plain.

DNA analysis. (Fig. 7) The neighbor-joining phylogenetic tree based on the COI mtDNA sequences of a set of TEP Halichoeres species, following the Kimura two-parameter model (K2P) generated by BOLD (Barcode of Life Database), shows relatively low divergences between species within complexes and deep divergences between more distant sets of species. The postion of H. sanchezi was recovered in a clade containing the two other members of the complex, i.e., H. salmofasciatus and H. melanotis (note that there are no sequences available for H. malpelo). The members of the complex are less than 3% divergent from each other: 2.4% between H. sanchezi and H. melanotis, 2.9% between H. sanchezi and H. salmofasciatus, and 2.1% between H. melanotis and H. salmofasciatus (all uncorrected pairwise distances). All other Halichoeres species are more than 13.5% different from the H. melanotis species complex.

Figure 7 Neighbor-joining phylogenetic tree of COI mtDNA sequences for the New World species of Halichoeres, including Halichoeres sanchezi n. sp.

Species, location, and GenBank accession number are indicated for each sequence. Bootstrap values are shown at branch points, the asterisks represent values below 40%. Halichoeres argus is used as an outgroup. The scale bar at left represents a 2% sequence difference.

Interestingly, another endemic wrasse, Halichoeres insularis Allen & Robertson, 1992, was discovered on a prior expedition to the Revillagigedo Archipelago in 1991. That species diverges by 7.6% in mtDNA COI sequence from its widespread mainland sister species, Halichoeres dispilus Gunther, 1864. The wide range in divergences of the putative endemic species in the Revillagigedo Archipelago, from 0.5% for Thalassoma virens Gilbert, 1890 (from Thalassoma purpureum Forsskal, 1775) up to 7.6% for H. insularis, makes it difficult to discern a pattern in phylogenetic age for the splitting of species in the region. Note the cytochrome-b sequence (GenBank accession number GU938863) for “H. insularis” used in Wainwright et al. (2018) is apparently a mislabeled sequence of H. salmofasciatus, leading to a mistaken assignment of H. insularis to the H. melanotis clade in the Bayesian timetree in their Fig. 2 and in Rocha, Pinheiro & Gasparini (2010): there were no tissues from Revillagigedo available to them at the time. An additional correction regarding the phylogenetic tree is that Halichoeres burekae Weaver & Rocha, 2007 as listed by Victor, Alfaro & Sorenson (2013) (GenBank accession number JN313704), is now identified as Halichoeres caudalis (Poey, 1860) and newly obtained sequences for H. burekae are used in the tree here.

Comparisons. At the time of prior descriptions, there were few or no underwater photographs of various phases of the H. melanotis species complex to compare with, thus the table of color-pattern differences in Allen & Robertson (2002; Table 2) is no longer accurate. An updated table of color pattern differences is presented here in Table 2. Notably, Bessudo & Lefevre (2017), in their guide to the fishes of Malpelo, follow the 2002 table describing IP H. malpelo as having neither the opercular-flap or caudal-peduncle spots and thus label their photographs of IP fish with the caudal spot as “H. salmofasciatus”, and those without as H. malpelo (and inaccurately listing both species for Malpelo Island). Recently, a set of additional underwater photographs have been taken of H. salmofasciatus on Cocos Island (Figs. 8 and 9), H. malpelo on Malpelo Island (Figs. 10 and 11), and H. melanotis from Panama and Baja California (Figs. 12–14). These photographs greatly expand the known color variations for these species. Halichoeres sanchezi is most similar in appearance to its southern island relatives. The IP adults of both H. salmofasciatus and H. malpelo closely resemble H. sanchezi, except they lack the mid-dorsal-fin spot that is diagnostic of IP adult H. sanchezi. Allen & Robertson (2002) reported that the spot is only present on juvenile H. salmofasciatus below about 20 mm SL and a diagnostic photograph from Cocos Island shows the loss of the spot as individuals get larger (Fig. 8) (juveniles of H. malpelo have not been documented). The underwater photographs of the three H. melanotis complex species show wide variation in the color patterns of IP fishes with varying intensities of red from candy-stripe red to brown and the opercular-flap spot and the caudal-peduncle spot sometimes faded or absent.

Table 2 Color patterns on juveniles, initial phase, and terminal phase of the four members of the H. melanotis species complex in the tropical eastern Pacific Ocean.

Species	Juvenile	Initial-phase	Terminal phase	
H. sanchezi, n.sp.	Mid-dorsal fin spot large, oblong, upper edge yellow, extends onto body scales. Lateral stripe red on white.	Mid-dorsal fin spot smaller, oblong, upper edge yellow, can extend onto body scales. Lateral stripe red on white.	Black blotch on base of caudal fin. Head vermilion, body orangish brown with dark scale outlines. White patch on upper abdomen. Opercular flap without distinct black spot.	
H. salmofasciatus	Mid-dorsal fin spot small and round, not onto body. Lateral stripe red on white.	No mid-dorsal-fin spot. Lateral stripe usually reddish on white or pink background, stripe sometimes dusky or dark or fading away posteriorly.	Black band at end of caudal fin. Body blue-greenish with dark scale outlines. Small blue spots and lines in band behind eye. Opercular flap usually with distinct black spot.	
H. malpelo	Undocumented	No mid-dorsal-fin spot. Lateral stripe usually reddish on white or pink background, stripe sometimes dusky or dark or fading away posteriorly.	Black band at end of caudal fin. Body blue-greenish with dark scale outlines. Small blue spots and lines in band behind eye. Opercular flap sometimes with distinct black spot.	
H. melanotis	Dorsal-fin spot usually absent, can be small, rarely large and yellow edged. Black lateral stripes on a bright yellow-gold body, less often on a white or brownish body; stripes can be broken or absent.	No mid-dorsal-fin spot. Color pale, brownish, greenish or salmon; usually no lateral stripe on body, but striped gold form can persist in IP. Often small spots and lines in band behind eye. Often patch on abdomen with bright white myomere outlines.	No black on caudal fin. Body green, blue, yellow or brown shades, no dark scale outlines or lateral stripe. Small blue spots and lines over operculum and cheek and thin white band on lower head. Opercular flap with distinct black spot.	

Figure 8 Halichoeres salmofasciatus, juveniles and IP, Cocos Island, Costa Rica.

(A) Juveniles. Photo credit: Allison Morgan Estape. (B) Small IPs. Photo credit: Ross Robertson. (C) IP adult. Photo credit: Allison Morgan Estape.

Figure 9 Halichoeres salmofasciatus, TP males, Cocos Island, Costa Rica.

(A) Photo credit: Allison Morgan Estape. (B) Photo credit: William Bensted-Smith. (C and D) Photo credit: Ross Robertson.

Figure 10 Halichoeres malpelo, IP adults, Malpelo Island, Colombia.

(A) Photo credit: Yves Lefevre, Biosphoto. (B) Photo credit: Yann Hubert, iStock. (C) Photo credit: Graham Edgar (D) Photo credit: Sandra Bessudo, Malpelo and Other Marine Ecosystems. (E) Photo credit: Ross Robertson.

Figure 11 Halichoeres malpelo, TP males, Malpelo Island, Colombia.

(A and C) Photo credit: Yves Lefevre, Biosphoto. (B) Photo credit: Graham Edgar.

Figure 12 Halichoeres melanotis, juveniles from Panama and Baja California, Mexico.

(A) Coiba, Panama. Photo credit: Allison Morgan Estape. (B) Perlas Islands, Panama. Photo credit: Allison Morgan Estape. (C and D) Coiba, Panama. Photo credit: Allison Morgan Estape. (E) Baja California. Photo credit: Leonardo Gonzalez, depositphotos.com.

Figure 13 Halichoeres melanotis, IP adults from Baja California, Mexico and Panama.

(A) Baja California, Mexico. (B) Perlas Islands, Panama. (C–E) Coiba, Panama. Photo credit: Allison Morgan Estape.

Figure 14 Halichoeres melanotis, TP males, from Baja California, Mexico and Panama.

(A) Baja California, Mexico. Photo credit: Carlos Estape. (B) Perlas Islands, Panama. Photo credit: Allison Morgan Estape. (C and D) Coiba, Panama. Photo credit: Allison Morgan Estape.

The juveniles and IP of mainland H. melanotis are especially variable in appearance, with multiple different versions of juvenile color patterns (Fig. 12). One is the classic golden-yellow, black-striped juvenile (leading to the common name Golden Wrasse) as illustrated in Allen & Robertson (1992, 1994): it has been photographed frequently both in Panama and Baja California. The golden juveniles can have a mid-dorsal-fin spot on very small individuals, but it is quickly lost with age. In contrast, recent photographs by Allison Morgan Estape and Carlos Estape document quite different versions of juveniles and small IP fish in Panama and Baja California, coexisting with golden juveniles. Some are mostly black-and-white versions of the golden form, but others show dark bars with a variety of colors and a prominent colorful opercular flap with black and yellow and blue, similar to the flap of TP males; some of these small individuals can have a large round mid-dorsal spot, edged on the upper portion with yellow but not extending partially onto body scales as in juveniles of H. sanchezi. Another juvenile H. melanotis from the Sea of Cortez shows a different color pattern but retains the opercular flap markings and has a large caudal-fin-base spot but no dorsal-fin spot (Fig. 12). Both juvenile and IP phases can sometimes have a black spot on the membrane of the first dorsal-fin spine, a feature of TP males of the H. melanotis complex. The significance of this extreme variation in color patterns in this species remains to be clarified.

Larger IP adult H. melanotis also have a wide variety of color patterns, from pale salmon with the same black spots on the opercular flap and caudal-fin base of the other members of the complex to shades of grey, yellow, green, or brown with the black opercular-flap spot the only prominent marking (Fig. 13). Others show various degrees of a colorful barred pattern, some with a prominent “abdominal window” marking, a reddish purple or grey patch with conspicuous white streaks of myotomal fascia (this marking occurs in some other labrids, including Halichoeres nebulosus (Valenciennes, 1839) and some razorfishes). Larger IP adults can also show some blue lines and spots anterior to the opercular flap, but less prominent than on TP males.

The TP male of H. sanchezi shows the most divergence from relatives, most conspicuously the large black blotch covering the posterior caudal peduncle and the proximal half of the caudal fin. In contrast, the TP males of H. salmofasciatus (Fig. 9) and H. malpelo (Fig. 11) have black covering the posterior end of the caudal fin. They have a cross-hatched scale pattern on a mostly uniform greenish blue background with a pale abdomen, a yellowish wash on the snout, a bright red iris, and most have a pattern of reticulated blue lines and spots within the band behind the eye. The opercular flap ranges from a black oval to an inconspicuous bluish patch. Some photographs of H. salmofasciatus show a broad dusky midlateral band, broader than the red stripe of the IP, with only a dusky area on the rear caudal fin and these may include transitional stages from IP females to TP males. When disturbed, all adult members of the complex can display a barred pattern.

In sharp contrast to H. sanchezi, the TP male of H. melanotis shows a wide range of color patterns (Fig. 14). Its most diagnostic feature is the absence of a dark caudal-fin marking as found on island species. The head and body can range from blue-green to yellowish to orange, the body can show cross-hatching with vertical blue lines or rows of spots on each scale or can be uniform. The opercular flap is usually a distinct black vertical oval and a pale band extends from the chin across the lower operculum. All TP males have some degree of a complex pattern of small blue spots and lines on the head behind the eye. Some individuals with mostly IP patterns can also show these head patterns, perhaps including transitional stages. When the dorsal fin is raised, there is a small black spot on the first dorsal-fin membrane, as is found on TP H. sanchezi.

It is unclear what differentiates the IP and TP phases of H. salmofasciatus and H. malpelo from each other. In view of the variation in markings within these species, the taxonomic status of the two species needs to be confirmed by additional documentation, as well as by molecular methods, which can assist in evaluating populations for species-level differences.

Remarks. The joint expedition in November 2022 to the Revillagigedo Archipelago pioneered a “gold standard” approach to documenting an inventory of the fish assemblage of an island by combining diagnostic underwater photographs of the full set of species encountered, targeted collection of the photographed subjects, on-deck photographing of the fresh specimens, tissue sampling for molecular analysis, and preservation of vouchers for museum collections. It brought together an international group of scientists (the authors, Carlos Sanchez, D. Ross Robertson, Michelle Gaither, and Fernando Duarte), along with a cadre of some of the most experienced regional tropical marine underwater photographers (in alphabetical order): Alasdair Dunlap-Smith, Allison Morgan Estape, Carlos Estape, Jeffrey Haines, Andres Hernandez, Ann S. Johnson, Keri Kenning, Serenity Mitchell, Ellie Place, Lee Richter, and Sara Richter (Fig. 15). The expedition resulted in one of the most comprehensive assessments of the fish fauna of a particular island to date, with more than 5,500 diagnostic underwater fish photographs of more than 152 species (Estape, 2023). The results are in preparation for publication and include a species list, a photographic inventory, a DNA-barcode coverage report and documentation of several new species to be described from the islands (D.R. Robertson, 2024, in preparation).

Figure 15 Members of the joint expedition to the Revillagigedo Archipelago in November 2022.

Left to right, front row: Ellie Place, Serenity Mitchell, Ann S. Johnson, Michelle Gaither, Ambar Camila Sanchez-Espinoza, Fernando Duarte, Keri Kenning, Benjamin Frable; back row: Andres Hernandez, Alasdair Dunlap-Smith, Jeff Haines, Carlos Sanchez, Benjamin Victor, Ross Robertson, William Ludt, Allison Morgan Estape, Carlos Estape, Sara Richter, and Lee Richter. Photo credit: Allison Morgan Estape.

A benefit of the comprehensive team approach to inventorying the fish fauna of an island is the opportunity for a detailed documentation of the discovery of a new species of fish. The novelty of this justifies a description of the process. Detailed videos and photographs of the collection dive, the process of locating, subduing and bagging the holotype, and the procedures following on board to process the specimens are archived at https://doi.org/10.5281/zenodo.8384735 for videos and https://doi.org/10.5281/zenodo.8384765 for photographs.

In this case, an unknown species of wrasse was fleetingly observed and photographed by Kreg Martin in 2013 (Fig. 3). Both the TP and IP were observed at a single location, Punta Tosca on Socorro, at 20 m depth, on a REEF survey expedition led by Christy Semmens. Kreg Martin notes the fish were evasive and actively avoided divers and were especially difficult to photograph. At the time, the photographs were shown to author BCV and D. Ross Robertson and they concluded that it may represent an endemic species in the H. melanotis complex. However, no specimen was forthcoming and until our expedition, the status of the unknown species remained unresolved.

One of the priorities of the joint expedition was finding and documenting lesser-known species, especially those new to science. We did not encounter the unknown wrasse at Socorro or Clarion. However, on the last stop of the trip, at San Benedicto on 30 November 2022, the first dive of the morning was at the “El Canyon” dive site, to visit the shark-cleaning station. On the way back from the deep location, divers passed over a coarse gravel bed interspersed with small rocks at about 20 m and Dr. Sanchez collected several small fishes, which included a single small IP of the new labrid species. On a subsequent dive targeting the same location, all divers focused on collecting, and a number of IP and juvenile wrasses were cornered and captured, not without difficulty since the fish sought refuge under rocks and then buried themselves in sand. Few TP males were observed, and one was cornered and lost. On a final dive, WBL managed to spear the previously lost TP male and WBL and BWF finally maneuvered the specimen into a bag and thus the holotype was firmly in hand (Fig. 16).

Figure 16 Underwater collection of the holotype of Halichoeres sanchezi n. sp. at San Benedicto Island, Revillagigedo Archipelago, Colima, Mexico.

Photo credit: Carlos Estape.

Endemism. The remoteness of the Revillagigedo Archipelago has certainly contributed to the development of endemism and the ratio of endemic species rivals that of any small oceanic island. Interestingly, the endemic fish species include both chaenopsids on the islands, two of the three known gobies, one of the two local Thalassoma species, T. virens (with a few records from Clipperton, and vagrants to Baja California in ENSO years (Victor et al., 2001), where they do not persist), and now two of the Halichoeres species. In comparison, the Galapagos Islands, Cocos Island, and Malpelo have no endemic Thalassoma species and fewer, if any, endemic Halichoeres species. They do match in having endemic chaenopsids and most of their gobies. A detailed exploration of Revillagigedo endemism will be reviewed in the upcoming inventory report of the fish fauna of the islands (Robertson et al., in preparation).

Conclusions

During an international effort to comprehensively inventory the ichthyofauna of the remote Revillagigedo Archipelago off the Pacific coast of Mexico, a new endemic species of wrasse (Labridae) was collected and documented by a team of underwater photographers collaborating with scientists. The new species, H. sanchezi, belongs to the H. melanotis species complex that is found in most parts of the region, and it diverges 2.4% in the mtDNA COI sequence from its nearest relative, H. melanotis from Panama, and 2.9% from H. salmofasciatus from Cocos Island, off Costa Rica. The juveniles and IP of the new species closely resemble those of H. salmofasciatus and H. malpelo (from Malpelo Island off Colombia), while the TP has a color pattern distinct from other relatives, most notably having a prominent black patch on the caudal peduncle and base of the caudal fin. The new species adds to the list of endemic fish species for the isolated archipelago and is an interesting case of island endemism in the region.

Supplemental Information

Supplemental Information 1 The FASTA sequences.

FASTA format nucleotide sequences for Halichoeres sanchezi (accession number OQ922018) and Halichoeres burekae (accession numbers OR588066 & OR588068).

We are especially grateful to Carlos Sanchez (UABCS) for initiating, arranging, facilitating, and leading the 2022 expedition, as well as creating an exceptionally friendly collegial atmosphere. Fernando Duarte, also from UABCS, is appreciated for his able assistance. D. Ross Robertson of the Smithsonian Tropical Research Institute provided unparalleled taxonomic expertise and his guide to SFTEP was the indispensable reference database to which to refer. Jack Grove and the East Pacific Corridor Alliance Foundation helped facilitate the expedition. Allison Morgan Estape was instrumental in organizing the logistics of getting so many people and their equipment in the same place at the same time. We are all indebted to Dora Sierra, the owner of the vessel Quino el Guardian, and the cooperative and patient crew and dive masters who made the trip possible (and safe). We thank William Bensted-Smith, Sandra Bessudo, Alasdair Dunlap-Smith, Graham Edgar, Allison Morgan Estape, Carlos Estape, Jeff Haines, Andres Hernandez, Yann Hubert, Kreg Martin, D. Ross Robertson, and Sara Richter for permission to use their photographs and Sofia Lopez, Christy Semmens, and Lourdes Vasquez-Yeomans for facilitating access to photographs and other assistance. Finally, the group of enthusiastic underwater photographers: Alasdair Dunlap-Smith, Allison Morgan Estape, Carlos Estape, Jeffrey Haines, Andres Hernandez, Ann S. Johnson, Keri Kenning, Serenity Mitchell, Ellie Place, Lee Richter, and Sara Richter, who provided much of the human energy for the project, need to be acknowledged for their unparalleled skill in finding fishes, creating superbly composed photographs (usually while upside down and rolling), and being endlessly upbeat, humorous, and passionate about fishes and marine biology.

Additional Information and Declarations

Competing Interests

Author Contributions

Animal Ethics

Field Study Permissions

DNA Deposition

Data Availability

New Species Registration

The authors declare that they have no competing interests.

Benjamin C. Victor conceived and designed the experiments, performed the experiments, analyzed the data, prepared figures and/or tables, authored or reviewed drafts of the article, and approved the final draft.

Benjamin W. Frable conceived and designed the experiments, performed the experiments, analyzed the data, prepared figures and/or tables, authored or reviewed drafts of the article, and approved the final draft.

William B. Ludt conceived and designed the experiments, performed the experiments, analyzed the data, prepared figures and/or tables, authored or reviewed drafts of the article, and approved the final draft.

The following information was supplied relating to ethical approvals (i.e., approving body and any reference numbers):

Collection methods approved by the Secretariat of Agriculture, Livestock, Rural Development, Fisheries and Food (SAGARPA) of the Federal Government of Mexico (PPF/DGOPA-A-099/22).

The following information was supplied relating to field study approvals (i.e., approving body and any reference numbers):

Collections were made under a permit to the Universidad Autonoma de Baja California Sur, issued by the Secretariat of Agriculture, Livestock, Rural Development, Fisheries and Food (SAGARPA) of the Federal Government of Mexico (PPF/DGOPA-A-099/22).

The following information was supplied regarding the deposition of DNA sequences:

The nucleotide sequences for Halichoeres sanchezi and Halichoeres burekae are available at GenBank: OQ922018, OR588066, OR588068.

The following information was supplied regarding data availability:

The CT scans are available at MorphoSource:

- Holotype DOI: 10.17602/M2/M570594.

- Paratype DOI: 10.17602/M2/M570600.

The following information was supplied regarding the registration of a newly described species:

Publication LSID: urn:lsid:zoobank.org:pub:0FF79EE8-EB5A-45B3-98CE-9A420881DAA0.

Halichoeres sanchezi species LSID: urn:lsid:zoobank.org:act:608591F4-2A1B-4271-9BB2-0F0B2BD2DC4E.

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
