# Peer review of "Halichoeres sanchezi n. sp., a new wrasse from the Revillagigedo Archipelago of Mexico, tropical eastern Pacific Ocean (Teleostei: Labridae)"

_PeerJ, doi:10.7717/peerj.16828_

## Round 0.1 · original submission · Major Revisions

The manuscript on the new species of wrasse fish from the Pacific Ocean is interesting. Three reviewers have given their valid comments to improve the manuscript. Please follow their comments carefully and revise accordingly.

·

Basic reporting

This paper describes a new, distinct species of New World Halichoeres from the H. melanotis species complex. While the description is justified and straight forward, the manuscript is unnecessarily long in some parts. There are 22 figures, many of which can be trimmed down to a more manageable number. Indeed, the figures aren’t cited in order, and many aren’t even cited in the main text. For example, in text citations for figures 13, 21, and 22 are missing. I’ve provided suggestions below on some ways to trim down text and figures. I have also included an annotated pdf for other minor comments not listed here.

The study puts a lot of emphasis on the collecting and sampling methods, but then the methods detailing the barcoding and molecular component of the study are a little bit lacking. For example, the manuscript explicitly mentions a pioneering approach of collecting and sampling, so it is a little bit strange that barcoding was done only for one specimen of the new species, given that there were 8 in the type series. In my opinion, the workflow of collecting, photographing, tissue sampling, and vouchering, is fairly common practice nowadays and is neither new, nor pioneering. Many expedition teams do this (the California Academy of Sciences, Bishop Museum, and Smithsonian for example). I do, however, agree that this should be the gold standard – so props to the authors for the detailed methodology.

M&Ms – Provide a little bit more info here. Where were the tissue samples collected? Fin? Muscle? From the holotype? One of the paratypes? Briefly mention the preservation method (e.g., tissue samples were collected from the right pelvic fin of the holotype and x paratypes, preserved in ethanol, and stored at -20C prior to extraction).

I would include a separate sub-section in the M&Ms describing the molecular component. Describe how the outgroups and comparative ingroup species were chosen. The tree figure (figure 12) is missing node support values (this should be given even in a NJ tree). The figure caption mentions comparative Halichoeres species of the new world, which is great, but this should be mentioned in the methods as justification for choosing comparative species. Perhaps like “the new species was compared with those belonging to the H. melanotis complex as well as other species in the New World”.

Lines 117-123: PCR protocols can be deleted entirely and summarized as a citation. E.g. “PCR protocol and conditions follow those described in xyz”.

Lines 131-140: This entire paragraph can be deleted in its entirety. You would assume that the new species description is in accordance with ICZN. There is no need to mention this, nor Zoobank. All of this is common practice that does not need to be stated.

Diagnosis, lines 161 and beyond: The dentition is not diagnostic of the species here so consider removing.

Description, Lines 178 and beyond: Presentation of caudal-fin rays is a little bit confusing here. “Segmented caudal-fin rays 17, upper two and lower three unbranched”. From the image of the holotype, I count 6+6 principal caudal fin rays, the upper and lowermost unbranched. The remaining rays should be procurrent rays. I think presenting caudal-fin rays as principal caudal fin rays and procurrent rays is less confusing than using segmented rays. The latter does not make a distinction between rays that articulate on the hypurals vs those that don’t. Segmentation of caudal-fin rays is likely also ontogenetic. You have x-rays and CT scans of this handy, so making the distinction between principal caudal and procurrent rays should be easy.

Description, line 182: “vertebrae 25”. Since you have x-rays and CT scans, why not present this as precaudal + caudal vertebrae, which, in Halichoeres, should be 10 + 15 (but please check). It provides more information and detail than total vertebral counts. Most labrids have 25 vertebrae but the number of precaudal + caudal differs between genera.

Description, lines 209 to 216: some of the values are presented as whole numbers. Consider including a decimal even if it’s a round number for consistency and in accordance with the methods (to 0.1 decimal place). i.e., 11.0 instead of 11.

Colour in life, line 217 and beyond: consider splitting this as two separate descriptions. Colour in life (TP) and colour in life (IP).

Line 286 and beyond: Inconsistent use of abbreviated genus names in this paragraph.
DNA analysis, lines 275 and beyond: Why not just use the word phylogenetic instead of phenetic? You are explicitly using DNA sequences to infer relationships for a given set of species. This is mentioned in the discussion of sister species and members of a closely related species group. So, use phylogenetic.

Comparisons, Lines 300 and beyond: consider making a table of coloration characters for the comparative species. They differ markedly in the various anatomical positions, so not only is this easy to do, but will also serve to be very useful and should be fairly easy to make.

Remarks, lines 386 and beyond: This paragraph seems a little bit unnecessary to me. Most, if not all can be deleted. Perhaps keep the line detailing that an upcoming report that is currently in preparation is to follow. That bit is good.

Figure 6 is in black and white. Is this intentional?

Figures 7, 8, and 9 are very similar. Consider deleting some or merging them all into a single plate.

Figure 6 and 11 can be combined with figures 2 and 3 into a single plate. Live, preserved, and fluorescent coloration of the male holotype and female paratype.

Figures 17 to 20: Consider merging into plates or removing some (i.e., Fig. 18).

Figures 21 and 22 are not necessary. Consider removing.

Experimental design

Comments above ^

Validity of the findings

The species is valid and is distinct.

Additional comments

See comments above and those in the annotated pdf file.

Reviewer 2 ·

Basic reporting

This is a very straightforward manuscript that describes a new species of wrasse, that is very well written.
As such, I only have very minor comments that I will provide at the end.
While this article is simple enough, in its format and casual style, that at times describes, uncharacteristically, how fish were caught, who was on the expedition and such, it is also incredibly thorough in its research for alternative possibilities, in describing evolutionary mechanisms, in assessing potential biogeographic implications, in correcting long standing errors in the literature, and finally in comparing what happens in Halichoeres and other, closely related genera such as Thalassoma.
In that respect, this paper is unique and incredibly valuable.
Understanding the mechanisms of endemicity in island populations of fishes is a big task, and this paper touches on many important topics.
I very much enjoyed reading it and was convinced by its conclusions.

Very minor details:
Line 50: it is stated that all species encountered were sampled. Is this really correct? I find this hard to believe, but will, of course, if the authors confirm it.
Line 53: videotape. Are you sure? was it not digitally recorded?

Experimental design

The authors have followed the main standards of species description. They photographed the specimens, and vouchered them, after doing their due diligence by doing meristics and morphometrics.
They used mtDNA markers, which is not the very best, but in this case perfectly adequate.

Validity of the findings

the papers provides very convincing evidence of their claims

Additional comments

very enjoyable read

·

Basic reporting

No comments

Experimental design

Regarding Methods, I strongly suggest the addition of a phrase as in commetary at line 98 of the ms.

Validity of the findings

Just an observation:
Unfortunately only one TP was collected, counted and mesaured. In a few characters (highlited in the attached reviewed PDF) the measurements seem to be somewhat odd, strongly differing from those of the paratypes. This kind of difference is not found in the descriptions of H. malpelo and insularis (Allen & Robertson 1992) and salmofasciatus (Allen and Robertson 2002)...
In other hand, the use of fluorescent techniques is remarkable, an innovation that clearly shows the effort of the authors to unveil all characters of the new species..

Additional comments

The description of new species is fundamental to Science. Knowing which organisms live in each environment, their behavior, role in the ecology of that place, theorize about how they evolved, can answer (and propose new) questions regarding our own survival.
This paper is a classic (already) among so many others, bringing to light the difficulties of collection and, even so, enabling the knowledge of a species fully adapted to the conditions of its environmental niche, including highlighting details about its elusive behavior.
Congratulations to the authros.

---

## Round 0.2 · Minor Revisions

Authors have done all corrections and a reviewer is suggesting to follow some minor corrections and hence I request the authors to follow those and revise it as minor revision.

·

Basic reporting

no comment

Experimental design

no comment

Validity of the findings

no comment

Additional comments

The authors have incorporated most, if not all of the suggest comments to a satisfactory level. I have provided some very minor additional edits, such as missing punctuations, the usage of em dashes instead of hyphens for range of values, and consistently of telegraphic language in the description. The authors may choose to ignore these if they wish. Overall a nice description to accompany a nice species!

·

Basic reporting

no comment

Experimental design

no comment

Validity of the findings

no comment

---

## Round 0.3 · accepted · Accept

Authors have made all corrections hence it can be accepted for publication. I wish authors to find more fish species that are new to science.